# mRNA Isoforms and Variants in Health and Disease

**DOI:** 10.3390/ijms26199356

**Published:** 2025-09-25

**Authors:** Sharmin Shila, Vinesh Dahiya, Charles Hisle, Elizabeth Bahadursingh, Ramkumar Thiyagarajan, Patrick E. Fields, M. A. Karim Rumi

**Affiliations:** 1Department of Pathology and Laboratory Medicine, University of Kansas Medical Center, Kansas City, KS 66160, USA; sharminshila.mib@gmail.com (S.S.); vinesh.dahiyampharm@gmail.com (V.D.); charles.hisle@ku.edu (C.H.); elizabeth.bahadursingh@ucdconnect.ie (E.B.); pfields@kumc.edu (P.E.F.); 2Department of Internal Medicine, University of Kansas Medical Center, Kansas City, KS 66160, USA; rthiyagarajan@kumc.edu

**Keywords:** gene expression, mRNA isoforms, cell differentiation, organogenesis, aging, molecular pathology, mRNA isoforms in diseases

## Abstract

Cellular gene expression varies in different physiological or pathological conditions. Conventional gene expression analysis assumes that each gene produces a single mRNA, which is not accurate. On average, genes express more than three mRNA isoforms. A particular cell type expresses different mRNA isoforms from a specific gene depending on its developmental or differentiation state. Different isoforms encode distinct proteins or noncoding regulatory RNAs, each with its own unique functions. Pathological states also alter the expression of transcript variants, which can either cause a disease or facilitate recovery. Still, the detection of mRNA isoforms or variants is not preferred to avoid complex analyses. As a result, gene expression studies often fail to identify the actual mRNA isoforms or variants associated with pathophysiology. In this article, we summarize the data on mRNA isoforms and disease-associated variants identified in various physiological and pathological conditions. These findings emphasize the importance of detecting mRNA isoforms for a better understanding of physiological or pathological conditions.

## 1. Introduction

mRNA isoforms are the various mature transcripts expressed from a single gene locus. The isoforms arise due to alternative transcription start sites (ATS), alternative splicing (ASP) of precursor mRNAs (pre-mRNAs), and alternative transcription termination (ATT) and alternative polyadenylation sites (APA). ATS and APA result in variation in the 5’-end and 3’-end of mRNA, respectively; however, ASP can alter any part of the mRNA [1]. Resulting mRNA isoforms may vary in coding sequence in the amino-terminal, carboxy-terminal, or other functional domains [2,3]. Enabling a single gene to encode multiple protein isoforms or noncoding RNAs(ncRNAs), the mRNA isoforms fine-tune cell signaling, differentiation, and adaptation to the environment [4,5].

Tissue-specific epigenetic regulators (ERs) and transcription factors (TFs) determine the ATS. RNA-binding proteins (RBPs), particularly splicing factors (SPFs), are involved in ASP. In contrast, transcription terminators and polyadenylation factors regulate the APA [4,5]. ATS, ASP and APA, as well as chemical modifications of mRNA molecules, regulate mRNA stability, localization, and translation efficiency [6]. Thus, many molecules are involved in transcriptional and post-transcriptional regulation of mRNA isoform expression. Tissue-specific differential expression of these regulators primarily determines the expression of a particular mRNA isoform in a cell. Expression of the mRNA isoforms is also tailored to distinct cellular niches, ensuring precise spatiotemporal control over gene expression [7]. The activities of retrotransposons and the quality control mechanisms of pre-RNA processing also influence these changes [8,9]. A summary of the mechanisms responsible for generating mRNA isoforms is shown in tabulated form (Appendix A).

Cells generate selective mRNA isoforms to adapt to dynamic physiological and pathological conditions [10,11]. This transcriptomic plasticity is essential for developmental programs, tissue-specific functions, and the functional decline during aging. In physiological contexts, mRNA isoforms orchestrate lineage specification, immune response modulation, and neuronal plasticity [12]. Accelerated transcription and splicing increase the chance of erroneous splicing events, affecting genes involved in cell proliferation, cell differentiation, and/or apoptosis [13]. However, perturbations in pre-RNA processing machinery due to epigenetic alterations, mutations in spliceosome components, or aberrant expression of RNA modifiers can generate pathogenic isoforms [14] (Figure 1). Dysregulation of pre-mRNA processing disrupts molecular homeostasis, contributing to various diseases [15,16,17]. However, those may also be generated in response to disease conditions.

In this article, we have termed the disease-associated mRNA isoforms as transcript variants. The influence of pathological transcript variants spans many systemic, genetic, metabolic, autoimmune, infectious diseases, and malignant diseases. The variants may play a pathological role or may be expressed to fight the disease conditions.

The role of mRNA isoforms or variants was minimally explored until the development of mRNA sequencing strategies. Advances in mRNA sequencing, notably long-read sequencing and single-cell transcriptomics, have unveiled previously unrecognized isoform-level alterations in physiological or pathological conditions [18]. In the following sections, we have discussed the expression of mRNA isoforms and variants in normal physiology, disease pathology, and their potential in translational research [19].

## 2. Physiological Role of mRNA Isoforms

Most mammalian genes express more than one mRNA isoform. mRNA isoforms of a gene diversify the encoded proteins and their functions. We have addressed the importance of mRNA isoforms in our previous publications [3,16,20]. In the following sections, we have discussed the physiological role of mRNA isoforms during embryonic development, organogenesis, normal functions, and aging.

### 2.1. Development and Organogenesis

The role of mRNA isoforms in development and organogenesis begins at fertilization, where maternal transcripts in oocytes guide early embryonic processes [21]. Then, the maternal-to-zygotic transition (MZT) marks a pivotal shift in gene regulation as zygotic genome activation (ZGA) replaces the maternal control [22]. During the transition, ASP generates mRNA isoforms essential for embryonic development [23]. In zebrafish embryos, genes like *Pou5f1* (*Oct4*), *Sall4*, and *Dnmt1* express isoforms with pluripotency and DNA methylation, essential for ZGA and early cell fate decisions [24]. Isoform changes in genes like *F11r* and *Magi1* alter coding proteins that facilitate cell–cell interactions during ZGA [25] (Figure 2A). RBPs such as CSDE1 undergo isoform switching during ZGA, through exon skipping that impacts maternal transcript clearance [26,27].

SPFs like SF3B1 and RBFOX2 also play crucial roles in generating isoforms that regulate pluripotency and lineage specification [28]. These splicing events are tightly coordinated with chromatin remodeling processes mediated by histone variants like H2A.Z, which destabilize nucleosomes to enable transcriptional activation [29,30]. Transcriptome-wide analyses in zebrafish and mouse embryos revealed the dynamic changes in isoforms that regulate tissue-specific gene expression [31]. As the embryo develops through gastrulation and organogenesis, waves of isoform switching correlate with the formation of major organs such as the heart and neural tube [32]. The ASP of *RBM20* and *NOVA2* transcripts generates isoforms essential for cardiac development and neural differentiation [33]. In addition, TFs such as SRF influence cardiac-specific mRNA isoform expression critical for myocardin-mediated developmental pathways [34,35,36]. mRNA isoforms of TFs like NR5A1, RUNX and BAD are involved in adrenal and gonadal development [37] (Figure 2B). NR5A1 isoforms are expressed in gonadal precursor cells and regulate the differentiation of the adrenal and gonadal tissues. Mutations or dysregulation of NR5A1 can lead to disorders of sex determination and adrenal insufficiency [38,39].

Epigenetic mechanisms further regulate the generation of mRNA isoforms during organogenesis. Histone acetyltransferase p300/CBP deposits H3K27 acetylation marks at active enhancers, facilitating tissue-specific gene activation [40,41]. Conversely, histone deacetylases (HDACs) compact chromatin to repress transcription in specific sites [42]. DNA methylation complements these processes by silencing non-essential genes, while demethylation activates enhancers critical for tissue identity [43,44]. Throughout organogenesis, APA generates mRNA isoforms with variable 3’ UTR lengths that impact post-transcriptional regulation [45]. For example, APA of *HMGA2* produces shorter isoforms with reduced regulatory complexity but increased translational efficiency, supporting rapid protein synthesis during development [46,47].

As the embryo matures into a fetus, mRNA isoforms continue to undergo complex developmental programs across tissues [48]. Aberrant expression or processing of these isoforms can lead to developmental abnormalities such as neurodevelopmental disorders or congenital defects [49]. Moreover, aberrant splicing events caused by synonymous or intronic variants can activate cryptic splice sites, leading to exon truncation or misregulation of developmental genes [50]. For instance, mutations affecting splicing regulators, such as FOXP4 or frameshift variants in HMGB1, disrupt normal gene expression. The frameshift variants in the HMGB1 protein have been shown to alter phase separation and nucleolar function, resulting in rare syndromes such as brachyphalangy, polydactyly, and tibial aplasia syndrome, underscoring the role of mRNA isoforms in proper organogenesis [48,51,52].

### 2.2. Physiological Functions

mRNA isoforms are fundamental to normal physiology after birth, contributing to tissue-specific gene expression, protein diversity, and cellular adaptation to environmental stimuli [53]. ATS, ASP and APA enable a single gene to produce mRNA isoforms, resulting in protein variants with distinct functions [16,54,55]. This process is essential for maintaining the specialized functions of different tissues and organs [56].

ASP, a major driver of transcript diversity, dynamically regulates gene expression across tissues. For instance, the CACNA1C gene, which is involved in cardiac function and blood pressure regulation, generates over 10,000 splice variants, with some isoforms specific to smooth muscle and others to cardiac tissue [57,58]. In the nervous system, splice variants of neurotransmitter receptors and ion channels play critical roles in synaptic plasticity and neuronal function [59]. The *Dscam* gene in Drosophila exemplifies an extreme diversity due to ASP, producing over 38,000 isoforms through the ASP of 95 variable exons that contribute to neural connectivity [60,61]. RBPs are key regulators of ASP and other post-transcriptional processes. RBPs such as SRSF1 and hnRNPs influence exon inclusion or exclusion, shaping protein structure and functionality critical for cell growth, differentiation, and metabolism. These proteins also regulate mRNA stability, localization, and translation efficiency, ensuring precise control of cellular processes in response to physiological changes [62]. RNA modifications further enhance regulatory complexity. For example, N6-methyladenosine (m6A) modifications influence mRNA structure, stability, splicing, export, translation, and decay [63,64]. During stress conditions like heat shock, m6A modifications in the 5’ UTR of HSP70 mRNA promote cap-independent translation to support cellular adaptation [65,66].

APA generates mRNA isoforms with variable 3’ UTRs, altering the elements for binding RBPs or miRNAs [67]. This mechanism enables tissue-specific gene expression and modulates protein abundance in response to environmental cues [68]. It has been reported that APA increases protein levels in skeletal and cardiac tissues while reducing neuronal gene expression through intronic cleavage [69]. The generation of mRNA isoforms is controlled by tissue-specific regulation of pre-mRNA processing. TBX20 isoforms are expressed selectively in tissues like the aorta, coronary artery, testis, pituitary gland, and heart [62]. Such specificity underscores the significance of mRNA isoforms in regulating gene expression across various organ systems.

### 2.3. Aging

The aging process profoundly impacts the generation of mRNA isoforms, driving changes in gene expression and cellular functions. As organisms age, ASP patterns shift, altering tissue-specific expression and function of critical genes [70]. Nearly 49,869 splicing events have been identified during human aging that can affect DNA repair, apoptosis, and RNA processing [71]. These changes are often accompanied by an imbalance in transcript length, with shorter transcripts linked to inflammation and functional decline, while longer transcripts are associated with increased lifespan [72].

Epigenetic changes play a significant role in altering mRNA isoform expression during the aging process. The loss of histone H3K36 methylation exposes cryptic promoters, leading to the production of non-canonical transcripts that may disrupt gene regulation and limit lifespan [73,74]. RBPs that regulate splicing, mRNA stability, and translation exhibit declining levels with aging [75]. Proteins like HuR and TIA-1 lose their ability to stabilize stress-response transcripts, weakening cellular repair mechanisms and exacerbating age-related damage [76]. Additionally, specific transcript variants like *Cdkn1a* variant 2, which encodes the P21CIP1/WAF1 protein, are a particular marker of aging and cellular senescence [77]. The elevated level of this isoform of *Cdkn1a* was observed in multiple tissues of aging mice, including the liver, adipose tissue, kidney, heart, and lung [78]. Post-transcriptional processes, including transcription elongation and ASP, remain central to the cellular landscape of aging. In the human brain alone, over 1174 exons exhibit differential expression with age, reflecting decreased ASP across multiple tissues, including blood, skin, muscle, bone, thymus, spleen, and adipose tissue [71]. Accelerated transcription elongation increases splicing and raises the likelihood of erroneous splicing, which is detrimental to cellular functions.

One hallmark of aging is the acceleration of RNA Pol II elongation speed across species and tissues [79]. This increased transcriptional speed disrupts the fidelity of RNA splicing, leading to reduced unspliced transcripts and elevated circular RNA formation [80,81]. Such changes alter isoform ratios and compromise protein activity, contributing to functional decline [82]. Genes involved in DNA repair and muscle homeostasis are particularly affected, with splicing errors resulting in aberrant isoforms that undergo nonsense-mediated decay [53]. It has been demonstrated that aging alters the splice variants of genes such as ESRRG and TET2 in skeletal muscle, which are essential for type-1 muscle fiber development and myogenic differentiation [82]. Aging also changes the splice variants of genes, which may cause muscle damage [82]. It has been found that lifespan-extending interventions, such as dietary restriction and reduced insulin-IGF signaling, can reverse those transcriptional and splicing alterations in skeletal muscle [83]. A summary of the physiological roles has been presented in the Appendix A.

Aging also activates retrotransposable elements, increasing genomic instability as these mobile genetic elements generate cDNAs through reverse transcription [8]. This activity can lead to genomic reinsertion and DNA damage, induction of new mRNA variant expression, and exacerbation of cellular dysfunction. Interestingly, lifespan-extending interventions like caloric restriction have been shown to suppress retrotransposon activity in aged mice, highlighting the connection between retrotransposon regulation and longevity [84,85]. Declines in RNA quality control mechanisms further compromise mRNA integrity during aging [86]. In organisms like *Caenorhabditis elegans*, age-associated reductions in mannosyltransferase ALGN-2 impair nonsense-mediated mRNA decay, allowing harmful splicing errors to accumulate. These breakdowns contribute to cellular dysfunction across aging tissues [87].

## 3. mRNA Transcript Variants in Disease Conditions

The generation of mRNA transcript variants plays a multifaceted role in disease conditions. Some transcript variants are formed in the host cells in response to the disease, while abnormal transcript variants can be responsible for disease pathogenesis. Pathogenic mRNA transcript variants may directly cause certain diseases, while others are implicated in disease progression or modulate therapeutic responses. For ease of understanding, we have described these transcript variants according to the types of human diseases they are associated with. A summary of the representative splicing mechanisms, related factors, and disease-specific variants has been included in the Appendix A.

### 3.1. Systemic Diseases

mRNA transcript variants play a crucial role in systemic diseases by diversifying protein functions and altering cellular processes [88]. The mRNA isoforms fine-tune the regulation of physiological functions, but their dysregulation disrupts the physiological balance and drives disease progression across multiple organs and systems [89].

#### 3.1.1. Neurological Diseases

The diversity of mRNA isoforms and variants is central to the disease mechanisms in neurological diseases [90]. Neurons require precise regulation of mRNA isoform expression to maintain their physiological functions. Disruptions in this process can lead to the development of diseases [91]. ASP generates multiple transcript variants that can influence synaptic function, neuronal differentiation, and plasticity [92]. Misregulation of RBPs, particularly SPFs, leads to aberrant isoforms (transcript variants), which are implicated in neurodevelopmental disorders such as autism spectrum disorder, intellectual disability, and schizophrenia [93]. Changes in the usage of ATS can result in altered protein expression and neuronal activity [94,95]. Dysregulation of APA further modulates mRNA stability and localization and can alter the expression of key neuronal proteins, contributing to disease progression [96]. In Alzheimer’s disease (AD), changes in the ratio of APP isoforms and variants are linked to plaque formation and neuronal damage [97]. ASP in MAPT pre-mRNAs generates tau variants that aggregate into neurofibrillary tangles, a hallmark of AD [97,98]. A recent study has identified read-through transcripts, such as *TOMM40-APOE*, at AD’s susceptibility loci [99]. Similarly, neuropsychiatric disorders are linked to novel mRNA variants of *ATG13* and *GATAD2A* genes, which alter the protein domains and play a pathogenic role [100].

#### 3.1.2. Cardiovascular Diseases

Hypertension highlights the role of mRNA variants in common cardiovascular diseases. Variants in genes like *KLF4* and *KLF5* affect vascular smooth muscle cell (VSMC) behavior, contributing to blood pressure regulation [101]. The *KLF5* variant rs9573096 promotes VSMC dysfunction, increasing hypertension risk, while elevated *KLF4* mRNA levels drive VSMC proliferation and worsen vascular injury [101,102]. In atherosclerosis, transcript length variations in genes like *MIA3* reduce VSMC proliferation and impair plaque stability, increasing cardiovascular risk [103,104]. In hypertension, particularly pulmonary arterial hypertension, the splicing of genes such as BMPR2 produces variants with varying activities, influencing disease severity and vascular remodeling [105]. Changes in polyadenylation and transcription start sites further modulate gene expression and vascular cell adaptation [106]. In cardiac failure, ASP in pre-mRNAs of sarcomeric genes, such as TTN, leads to the expression of altered proteins that affect myocardial stiffness and contraction [107]. Similarly, the ASP of *TNNT2* and *MYH7* pre-mRNAs alters proteins essential for cardiac muscle function [107]. During myocardial infarction, ASP variants promote cell survival and repair mechanisms, while circular RNAs derived from ASP transcripts regulate gene expression [108]. In cardiac hypertrophy, SPFs like RBM20 control the mRNA transcript variant changes that impair cardiac properties [109,110].

#### 3.1.3. Respiratory Diseases

Chronic respiratory diseases such as asthma and chronic obstructive pulmonary disease (COPD) exhibit significant dysregulation in mRNA isoforms [111]. In asthma, ASP of the *ADAM33* pre-mRNA generates variants that lack catalytic domains required for metalloprotease activity, thereby affecting airway remodeling [111]. Variants of *TGFBR1* and *SMAD3*, the key regulators of TGFβ signaling, contribute to airway inflammation and fibrosis by altering immune cell regulation [112,113]. COPD is characterized by frequent exon skipping and intron retention in severe cases, resulting in protein-coding transcript variants that impact lung remodeling and inflammation [114]. The spliced variants of the *SERPINA1* gene reduce AAT levels, leading to tissue damage and emphysema [115,116].

#### 3.1.4. Gastrointestinal Diseases

In the digestive system, the role of mRNA transcript variants is similarly critical, though less studied compared to neurological or cardio-respiratory diseases. Cells of the gastrointestinal tract, including enterocytes, goblet cells, and enteroendocrine cells, rely on tightly regulated gene expression for proper function, barrier integrity, and immune response [117]. The dynamic landscape of mRNA transcript variants underlies the complexity and variability of disease phenotypes and responses to environmental or genetic insults [118,119]. ASP and APA generate transcript variants of digestive enzymes, transporters, and signaling molecules [120]. ASP of transcripts that encode proteins for nutrient absorption or gut barrier maintenance can lead to malabsorption syndromes or inflammatory bowel diseases [121,122]. ATS and APA can influence the abundance and stability of transcripts encoding digestive enzymes or receptors, affecting overall digestive function and contributing to celiac disease, Crohn’s disease, and ulcerative colitis [123].

#### 3.1.5. Genitourinary Diseases

mRNA transcript variants play crucial roles in genitourinary diseases by modulating protein function and cellular processes. In the kidney, the transcript variants are differentially expressed in compartments like the glomerular and tubulo-interstitial regions, where splicing regulators such as ESRP1, ESRP2 and RBFOX2 govern critical developmental transitions like the mesenchymal–epithelial transition [124,125]. ASP contributes to disorders such as Wilms’ tumor, with distinct transcript variants of genes like *FGFR2* and *ARHGEF10L* affecting kidney functions [126,127]. In the ovary and testis, regulation of transcript variants influences cellular differentiation, spermatogenesis, follicular development, hormone responsiveness, and reproductive functions [128,129].

#### 3.1.6. Musculoskeletal Diseases

Transcript variants influence various musculoskeletal diseases by affecting critical enzymes and receptors. In vitamin D-dependent rickets, mutations in the *CYP27B1* gene reduce the production of active vitamin D, disrupting calcium balance and bone mineralization [130,131]. The ASP of CYP27B1 pre-mRNA generates variants of the vitamin D receptor that affect bone development [132]. Variations in SPF, such as hnRNPC, connect vitamin D signaling to ASP, further contributing to the disease [133]. In congenital multi-minicore myopathy, mutations in muscle-specific exons of the *FXR1* gene produce altered isoforms that compromise muscle fiber structure and function, causing muscle weakness [134,135]. Duchenne muscular dystrophy involves ATS and ASP to generate distinct dystrophin transcript variants that result in deficient or abnormal dystrophin-associated glycoprotein complex, leading to progressive muscle degeneration and cognitive impairments [136,137]. Mis-splicing of pre-mRNAs coding the proteins critical for neuromuscular junction formation impairs synaptic signaling between nerves and muscles in congenital myasthenic syndromes [138]. In spinal muscular atrophy, ASP reduces the survival motor neuron proteins, causing motor neuron loss and muscle atrophy [139,140]. Amyotrophic lateral sclerosis involves splicing abnormalities that disrupt RBP and cytoskeletal regulators, impair motor neuron function, and lead to neurodegeneration, muscle weakness, and paralysis [141,142].

### 3.2. Metabolic Diseases

In diabetes mellitus, altered mRNA isoforms (variants) impair insulin signaling, β-cell function, and glucose homeostasis [143,144]. RBPs that regulate mRNA processing are dysregulated, causing abnormal mRNA transcript variants [145,146]. Aberrant mRNA transcript variants impair β-cells maturation and insulin production. Additionally, changes in transcript variants affect the expression and activity of key metabolic genes, resulting in insulin resistance and exacerbating the disease [147,148]. It has been shown that increased levels of HuR stabilize the mRNAs of pro-inflammatory cytokines, contributing to diabetic nephropathy and retinopathy [149].

Obesity and metabolic syndrome also feature splicing alterations in genes regulating adipogenesis, energy homeostasis, and inflammation [150]. Isoforms of PPARs modulate lipid metabolism and insulin responsiveness, impacting disease development [151]. Splicing defects in mitochondrial genes impair energy metabolism, contributing to metabolic dysfunction [152]. In non-alcoholic fatty liver disease, altered transcript variants of specific enzymes and transporters impair lipid metabolism, and oxidative stress worsens liver steatosis and inflammation [153]. Inherited metabolic disorders such as phenylketonuria result from ASP of PAH pre-mRNA that reduces the activity and causes accumulation of toxic metabolites [154,155].

### 3.3. Genetic Diseases

mRNA transcript variants can play a pathogenic role in many genetic diseases, including Down Syndrome, cystic fibrosis, and thalassemia. In Down syndrome, the presence of an extra copy of chromosome 21 leads to widespread changes in gene expression, including altered ratios of neuronal subtypes and shifts in transcript variant expression for many genes [156,157]. Genes like *APP* and *BIN1* show distinct transcript variants switching in Down syndrome brains due to ASP [157,158]. The overexpression of splicing regulators such as DYRK1A, located on chromosome 21, drives aberrant splicing of the entire transcriptome [159]. These changes in splicing and transcript switching in the brain are implicated in the cognitive impairments and early-onset Alzheimer’s pathology seen in Down syndrome [157,160]. Tay-Sachs disease, a neurodegenerative condition, results from *HEXA* gene mutations, which disrupt mRNA processing, leading to loss of the mRNA or expression of nonfunctional protein, causing neurodegeneration [161,162].

In cystic fibrosis, mutations in the *CFTR* gene can affect splicing or polyadenylation—producing unstable or nonfunctional mRNA transcript variants, which impair the chloride channel function and contribute to the disease [163,164]. Thalassemia, particularly β-thalassemia, is caused by mutations that affect the normal splicing of β-globin pre-mRNA. Mutations at splice sites or within introns can activate cryptic splice sites, leading to aberrantly spliced mRNAs that are either degraded or translated into nonfunctional proteins [165,166]. Some mutations disrupt transcription initiation or termination, further reducing the amount of β-globin mRNA [167]. The interplay between ASP and polyadenylation is crucial here, as aberrant splicing can either expose or mask polyadenylation signals, thereby altering the stability and translation of the resulting transcript variants [168].

### 3.4. Autoimmune Diseases

Aberrant mRNA transcript variants can cause autoimmune diseases by generating proteins that disrupt immune homeostasis and tolerance [169,170]. In systemic lupus erythematosus (SLE), dysregulated splicing of *IRF5* and *CTLA4* pre-mRNAs leads to the expression of pro-inflammatory transcript variants and abnormal T-cell activation [171,172]. In rheumatoid arthritis (RA), splice variants of *CD44* and *TNFRSF1B* pre-mRNAs contribute to leukocyte migration and sustained inflammatory signaling, while nonfunctional *SIGLEC10* variants impair the anti-inflammatory responses [173]. In Sjögren’s syndrome, ASP of *OAS1* pre-mRNA results in the production of transcript variants, such as p42, p44 and p48, rather than the canonical p46 isoform [174]. The transcript variants impair responsiveness to IFN stimulation and contribute to the production of autoantibodies, leading to chronic inflammation [175,176,177]. In multiple sclerosis (MS), dysregulation of RNA splicing leads to the expression of autoantigens from the genes that generally maintain myelination of axons [178]. These alterations exacerbate immune-mediated damage to the central nervous system, contributing to the pathogenesis of MS [179]. In myasthenia gravis, defective *CHRNA1* pre-mRNA splicing encodes proteins that disrupt the assembly of acetylcholine receptors, impairing neuromuscular signaling and leading to muscle weakness [180]. Graves’ disease, an autoimmune thyroid disease, highlights the role of ASP in thyroid dysfunction. The *TSHR* gene produces truncated variants, such as ST4 and ST5, that impair central tolerance and give rise to autoreactive T cells to attack the cells expressing TSHR [181,182]. The production of autoantibodies against the TSHR affects TSH-signaling and leads to hyperthyroidism. ASP of other thyroid function-related genes, like *TPO*, further enhances autoantigenicity [181,183].

### 3.5. Infectious Diseases

Host and pathogen interaction modulates cellular mechanisms resulting in mRNA transcript variants that play a key role in infectious diseases, influencing host–pathogen interactions, immune responses, and disease progression [184,185,186]. The dynamic interplay and expression of transcript variants may contribute to the inhibition or promotion of the disease. In viral infections, expression of mRNA isoforms or variants benefits the pathogens by enhancing the replication and survival of the virus. Viral mRNA structure, including its 5′ and 3′ ends, is critical for evading host defenses and ensuring robust viral protein production [187,188].

HIV-induced global changes in host mRNA isoforms may disrupt genes involved in immune responses, cell death, and cell cycle regulation, impairing immune cell function, promoting chronic inflammation, and supporting viral persistence [189,190,191,192,193]. HIV infection-induced variants, including ASP and ATS, affect genes such as *CCR5*—a critical receptor for HIV entry [194]. Specific *CCR5* mRNA variants can increase receptor levels on immune cells, raising susceptibility to HIV infection and accelerating disease progression [195]. It also alters host genes like *CCNT1* and *RUNX1* to disrupt immune function [196,197,198]. Reduced expression of *RUNX1b* and *RUNX1c* variants that inhibit viral replication increases viral titer [199,200]. HIV-1 also promotes exon 7 skipping in *CCNT1*, suppressing transcriptional activation and maintaining latency in infected CD4 + T cells [197,198].

Among other viral infections, HBV is known to produce a spliced viral RNA variant, *SP1* RNA, which dampens inflammatory responses in the host to evade immune detection and establish a chronic infection [201,202]. In influenza infection, mRNA variant diversity—shaped by ATS, ASP and APA—plays key roles in both viral and host biology [203]. The influenza virus exploits host mRNA processing by “cap-snatching”. It uses the resulting protein variants to enhance viral adaptability and infectivity [204]. For example, EBV interferes with the host splicing machinery to skip exon 11 in the *MPPE1* gene, expressing an *MPPE1* variant that contributes to EBV-related tumorigenesis [205].

Bacterial infections exploit ASP to manipulate host defenses. Host response to infections changes the expression of mRNA variants to impair or augment immune response [206]. *Mycobacterium tuberculosis* (MTB) promotes exon skipping in the RAB8B gene to produce truncated variants that impair macrophage autophagy, allowing intracellular bacterial survival [207]. MTB further alters the splicing of the *IL-12Rβ* mRNA transcript to generate a shorter variant (IL-12Rβ1ΔTM) that enhances dendritic cell migration and MTB-specific T cell activation, promoting bacterial survival while modulating immune responses [208,209]. Other bacteria, like *Listeria monocytogenes*, influence the host splicing machinery through toxins like Listeriolysin O. This toxin induces the ASP of the cold-inducible RBP (CIRBP), resulting in two isoforms with opposing actions. While CIRBP-201 reduces intracellular bacterial load by promoting immune responses, CIRBP-210 facilitates bacterial survival by supporting stress-related pathways [210,211].

mRNA transcript variants in infectious fungal diseases, such as ringworm, arise from ASP, notably intron retention, in dermatophytes like *Trichophyton rubrum* [212]. These different mRNA variants can encode protein forms that influence fungal virulence and adaptation to the human host, particularly by altering the expression of invasion-related enzymes like keratin-degrading peptidases [213]. Parasitic infections also leverage mRNA variants regulation to disrupt the host cellular functions. In trypanosomiasis, skipping exon 15 of the *HDAC7* gene generates variants that inhibit host cell cycle pathways, promoting parasite survival and persistence [214,215].

### 3.6. Benign and Malignant Tumors

ASP and aberrant splicing of pre-mRNAs of oncogenes or tumor suppressor genes can be linked to tumor progression, metastasis, and resistance to chemo- or radiotherapy [216,217,218,219]. Differential expression of oncogenic mRNA variants may lead to differentiation of benign from malignant tumors [220]. The changes are driven by dysregulated oncogenic signaling pathways such as RAS/RAF/ERK, FOS/AP1 or PI3K/AKT/mTOR, which modify splicing factor activity at transcriptional or post-translational levels [221,222] (Figure 3A1). RAF1, a proto-oncogene in the MAPK/ERK pathway, generates transcript variants via ASP, ATS, or APA [223]. In lung cancer, these variants can enhance oncogenic signaling. FOS, a proto-oncogene that encodes a component of the AP-1, generates multiple transcript variants (Figure 3A2). Increased FOS levels or functions were found in uterine cancer [224]. ASP of *BCl2L1* pre-mRNA produces anti-apoptotic *BCL-XL* variants that protect cancer cells from programmed cell death [225]. These mechanisms are crucial for metabolic reprogramming and immune evasion in cancer [226]. Like upregulated oncogenes, downregulated tumor suppressor genes also contribute to cancer progression [227]. GLTSCR2, also known as NOP53, is a tumor suppressor gene that regulates cell growth, cell cycle progression, and stress responses, partly through controlling *MYC* expression in retinoblastoma (Figure 3B1) [228,229]. In skin cancer, abnormal transcript variants can change the expression, stability, or function of EMP3 (Figure 3B2).

A tumor-specific variant of the *EGFR* gene, *EGFRvIII*, is expressed in glioblastoma and promotes cell proliferation and resistance to therapies [230]. Notably absent in normal tissues, this transcript variant represents a promising therapeutic target due to its specificity for malignant cells [231,232]. In brain cancer, the ACY1 transcript produces multiple variants (Figure 4A). Abnormal splicing may create ACY1 isoforms that fail to regulate the cell cycle, resist apoptosis, and alter neural metabolism in ways that favor tumor growth and invasion [233]. Similarly, SRP19 pre-mRNAs also form multiple transcript variants in breast cancer (Figure 4B), and CDK5 produces numerous transcript variants in liver cancers (Figure 4C). Increased CDK5 activity phosphorylates TPX2, supports EMT, and activates oncogenic pathways such as mTORC1 and HIF-1α [234,235]. Cancer-specific variants of CDKN1A (p21) mislocalize it from the nucleus to the cytoplasm and impair its tumor-suppressive function in lung cancer [236] (Figure 4D). Similarly, dysregulated SMS variants can increase the growth and metastasis of prostate cancers [237] (Figure 4E).

In prostate cancer, variants of ADORA2B and COA1 also promote tumor progression [238,239]. Similarly, in prostate cancer, the *TMPRSS2-ERG* fusion transcript increases oncogenic potential [240]. This fusion protein induced by androgen signaling leads to ERG proto-oncogene expression, contributing to aggressive disease phenotypes [241,242]. In bladder cancer, isoforms such as the p63 variant ΔNp63α promote tumor invasion and metastasis by regulating epithelial–mesenchymal transition through microRNA control [243]. Germline mutations of the 3’ UTR of *CDKN2A* disrupt RBP binding, destabilizing mRNA and increasing the risk of melanoma [244]. For example, germline variants in the 3′ UTR of the *CDKN2A* gene, such as 500 C to G and 540 C to T, have been associated with increased melanoma risk and poorer prognosis by disrupting RBP binding and destabilizing mRNAs [245]. A summary of the alternatively spliced genes, isoforms, and their functions has been shown in the Appendix A.

Recent studies illustrate the emerging role of transcript variants in immune evasion of cancer cells and use them as therapeutic targets. Enrich the disease sections with findings from the last three years that demonstrate translational potential—highlighting the newest knowledge on isoform variants in tumor immune escape, which is of key clinical relevance. A newly identified isoform, PD-1IR2, derived from intron retention in the PDCD1 gene, has been shown to impair CD8^+^ T cell function. PD-1IR2 expression is induced upon T cell activation and regulated by the RNA-binding protein hnRNPLL. Functionally, PD-1IR2 suppresses T cell proliferation, cytokine secretion, and tumor cell killing, thereby promoting immune escape. In vivo models demonstrate that PD-1IR2-expressing T cells exhibit resistance to anti-PD-L1 therapy, positioning this isoform as a potential immune checkpoint and therapeutic target [246]. Alternative splicing of immune modulatory receptors in T lymphocytes has also emerged as a dynamic and targetable mechanism. Isoform-specific expression patterns, such as those of CD45, define T cell states and influence responsiveness to immunotherapy. Advances in long-read sequencing and chemically stabilized oligonucleotides have enabled precise manipulation of splicing events, allowing for the enhancement or silencing of specific isoforms to boost anticancer immunity [247]. Moreover, aberrant splicing in tumor cells can generate neoantigens—novel peptides presented via MHC molecules—that are absent in normal tissues. These isoform-derived neoantigens offer high specificity for malignant cells and are being explored as targets for personalized cancer vaccines and adoptive T cell therapies [248].

Studies reveal that cancer-specific splice variants play a central role in immune escape and therapy resistance. Novel isoforms of immune checkpoints, such as PD-1^28, suppress T-cell proliferation and cytotoxicity, enabling tumor growth [249]. PD-L1 splice variants lacking the transmembrane domain, such as PD-L1∆Ex5, are elevated in tumors and circulation, acting as decoys that reduce the efficacy of PD-L1 inhibitors [250]. Similarly, soluble CTLA-4 variants competitively bind therapeutic antibodies, promoting resistance to checkpoint blockade. Pharmacological inhibition of splicing regulators, including PRMT5 and RBM39, enhances antigen presentation, boosts T-cell infiltration, and induces tumor-specific neoantigens [251,252].

## 4. Translational Impact of mRNA Isoforms and Variants

The mRNA isoforms and disease-specific transcript variants offer a tremendous opportunity for translational research. As isoforms and variants play diverse roles in development, organogenesis, aging, and disease pathogenesis, they provide critical targets for clinical applications [29]. Long-read single-cell sequencing has revealed previously undetected variants, improving our understanding of cellular heterogeneity and disease mechanisms [253]. Advances in RNA sequencing techniques have enhanced our ability to detect the mRNA transcript variants, revealing previously unannotated transcripts that refine our understanding of gene regulation and disease mechanisms [20,254]. Large-scale transcript variants identified through advanced techniques like full-length ribosome–nascent chain complex sequencing have revealed novel neoepitopes derived from unannotated proteins [255]. It has been demonstrated that identifying isoforms enables the knowledge of how genes are expressed under physiological conditions and their contribution to disease [256]. Thus, studies on mRNA isoforms and variants have revolutionized translational research by uncovering the intricate regulatory mechanisms underlying disease pathogenesis [257].

In diagnostics, transcript variants have proven invaluable for distinguishing disease states. For example, specific isoforms, such as *CLU2*, in thyroid cancer can differentiate between malignant and benign tumors, enabling more accurate diagnoses and guiding preoperative decisions. Isoform analysis enhances diagnosis and prognosis, as isoform-level expression mediates significant cancer risk heritability and serves as an early detection biomarker [258,259]. It has been shown that transcriptomic profiling in breast cancer has also identified isoforms overexpressed in malignant cells, providing biomarkers for predicting disease progression and developing therapeutic strategies [260].

Therapeutically, targeting mRNA transcript variants has emerged as a promising approach for addressing genetic disorders and certain types of cancers. Splice-site-directed oligonucleotides and antisense therapies are developed to correct aberrant splicing and inhibit the expression of pathogenic transcripts [261]. Recently, spliceosome modulators targeting *SF3B1* mutations have been utilized in the treatment of hematological malignancies and solid tumors. In precision therapy, targeting specific isoforms or cancer driver exons may improve treatment efficacy and support personalized regimens [262,263]. Isoform-focused drug development identifies selective therapeutic targets, offering new avenues for intervention. FDA-approved therapies utilize these technologies to modulate RNA stability and translation, offering precision medicine [264,265,266]. These neoepitopes expand the repertoire of potential targets for personalized cancer vaccines, providing hope for more effective treatments across diverse tumor types [267,268].

Transcript variants also play a crucial role in pharmacogenomics, influencing drug metabolism and response. Variations in UTRs can alter mRNA stability and translational efficiency through interactions with miRNAs or RBPs, impacting how the body processes drugs [269]. Variations in drug-metabolizing enzymes like CYP3A4 affect statin dose requirements, underscoring the importance of transcript variant analysis in optimizing therapeutic outcomes [270].

## 5. Detection of Full-Length mRNA Isoforms and Variants

We have described the potential techniques for the detection of mRNA isoforms and variants [20]. However, recent advancements have significantly improved the efficiency of detection and understanding of full-length mRNA isoforms and disease-associated transcript variants. Full-length single-cell transcriptomics methods, such as Smart-seq3xpress, have been miniaturized to increase throughput while maintaining transcript completeness and detection sensitivity [271]. These methods enable the detection of both common and rare cell types across various tissues [271]. Similarly, FLASH-seq offers a streamlined, high-sensitivity alternative to traditional protocols, reducing hands-on time and reagent use, and facilitating automation. This approach is particularly efficient for capturing gene-body coverage and detecting alternative splicing events [272]. In the realm of single-cell long-read sequencing, HIT-scISOseq has demonstrated the ability to generate over 10 million high-accuracy long reads in a single sequencing run, significantly improving throughput and enabling detailed isoform analysis across diverse cell types [273]. ScISOr-Seq2 further advances this field by allowing the investigation of full-length isoforms across various brain regions, cell subtypes, and developmental time points, revealing widespread isoform variability and regulatory mechanisms [274]. On the computational front, Splice Transformer utilizes deep learning to predict tissue-specific RNA splicing alterations linked to human diseases, outperforming previous methods and enhancing variant interpretation [10]. Splam, another deep learning-based tool, improves splice site prediction by focusing on a shorter, biologically relevant flanking context, offering greater accuracy and efficiency compared to existing models [275].

## 6. Conclusions

Multiple mRNA isoforms can be expressed from a single gene, which vary in their 5’ end, 3’ end, or internal coding sequences. Thus, a single gene can express multiple proteins with diverse functional domains. The transcript variants may be expressed in varying quantities in different cell types or within the same cell type under various physiological or pathological conditions (Figure 5). The mRNA isoforms and the transcript variants can play diverse roles in cell differentiation, organogenesis, and aging processes. The mRNA isoforms or the transcript variants can also impact disease pathogenesis, progression, or response to therapeutics (Figure 5). This article discusses published reports on the physiological and pathological roles of mRNA transcript variants. These studies emphasize the importance of examining mRNA transcript variants. We anticipate that future studies using long-read mRNA sequencing will improve the identification of mRNA isoforms and their role in health and disease.

## Figures and Tables

**Figure 1 ijms-26-09356-f001:**
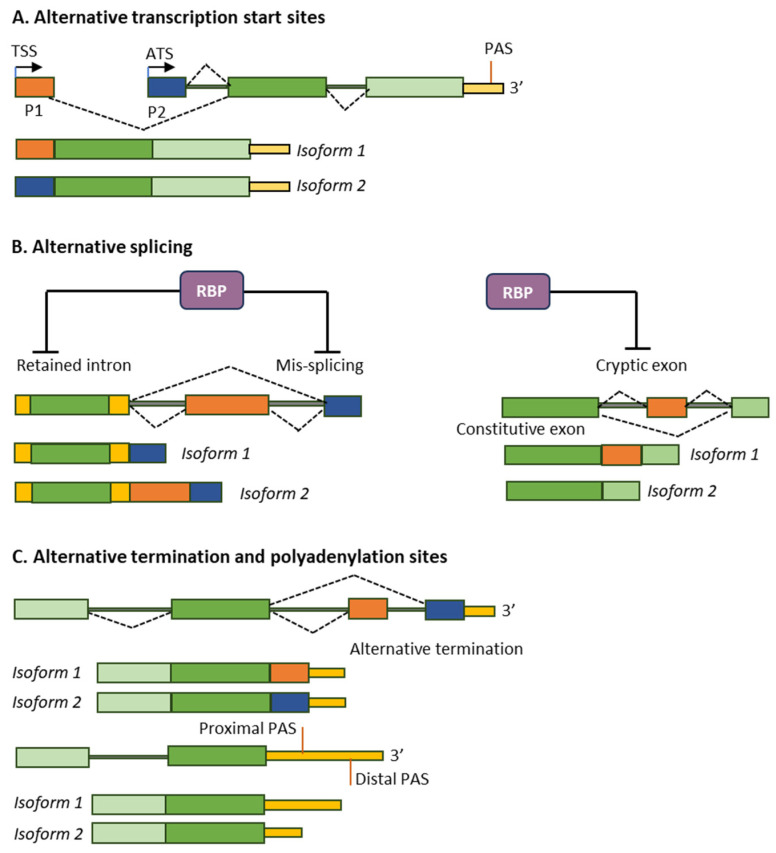
Dysregulation of RNA processing in diseases. In pathological conditions, dysregulation in pre-mRNA processing can occur at all steps, including ATS (**A**), ASP (**B**), and APA (**C**), resulting in transcript variants. Disease-related ASP can result from defects in the core splicing machinery or the dysfunction of specific RNA-binding proteins. The disease-associated transcript variants encode proteins that either aggravate a disease or help combat the pathological condition. Various exons are shown in different colors.

**Figure 2 ijms-26-09356-f002:**
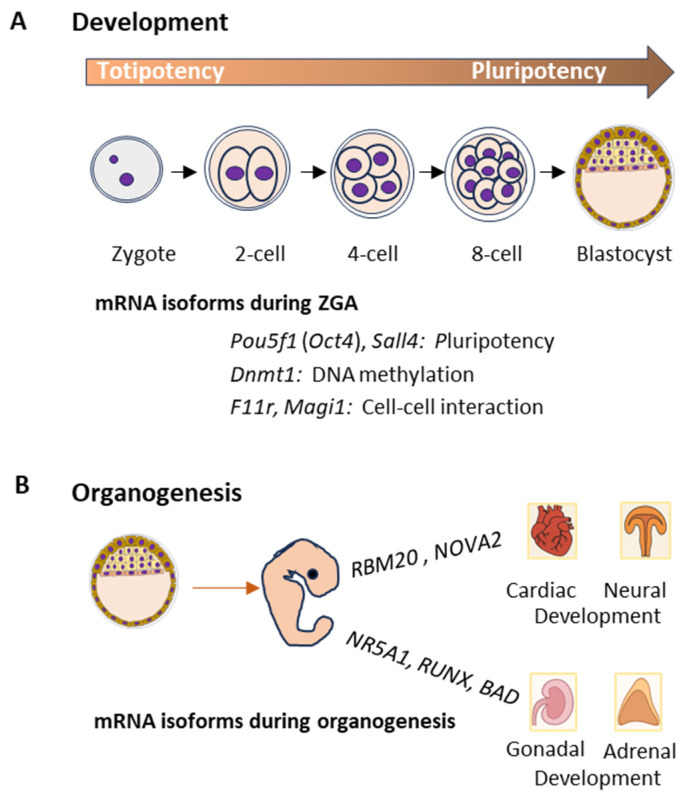
mRNA isoforms during embryonic development and organogenesis. A schematic presentation of mRNA isoforms during early embryonic stages from zygote to blastocyst shows the loss of totipotency and acquisition of pluripotency (**A**), followed by germ layer specification. (**B**) mRNA isoforms also regulate organogenesis, including the heart, kidney, nervous system, and lungs.

**Figure 3 ijms-26-09356-f003:**
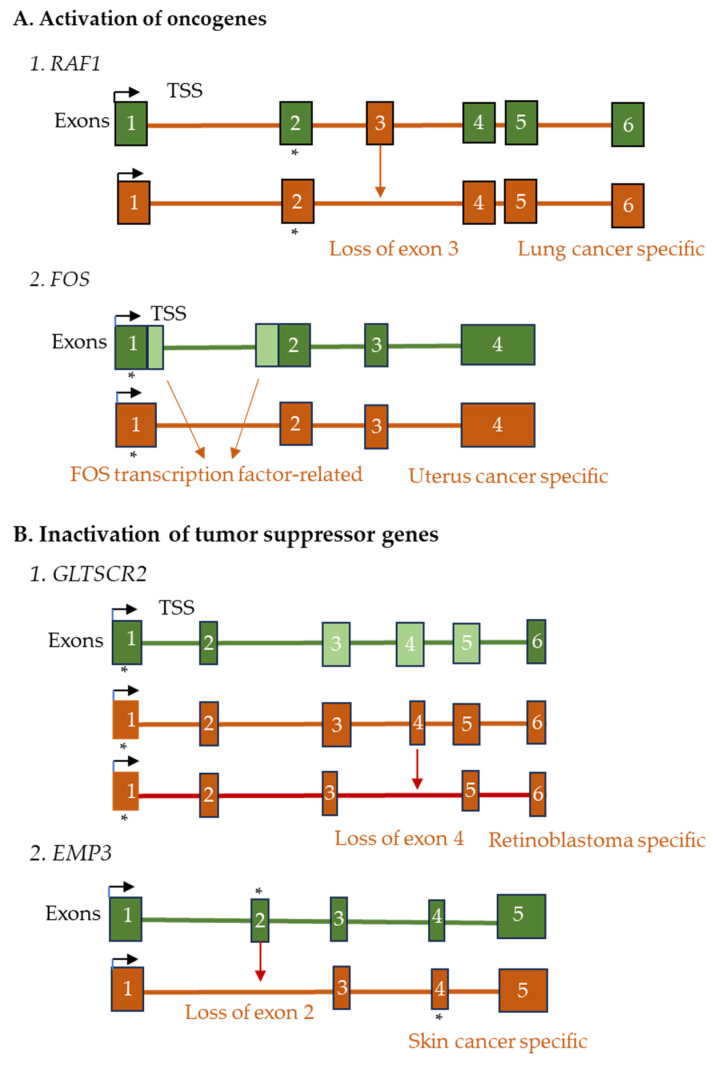
A schematic representation of cancer-specific alternative gene splicing. (**A**) Activation of oncogenes and (**B**) Inactivation of tumor suppressor genes. The ASP of RAF1 generates a lung cancer-specific transcript, whereas the ASP of FOS produces a uterine cancer-specific transcript. Tumor suppressor GLTSCR2 is alternatively spliced to produce two retinoblastoma-specific transcript variants and EMP3 to generate a skin cancer-specific variant. Deleted exons are shown with dark brown arrows, and start codons with * marks. The exons are numbered 1–6.

**Figure 4 ijms-26-09356-f004:**
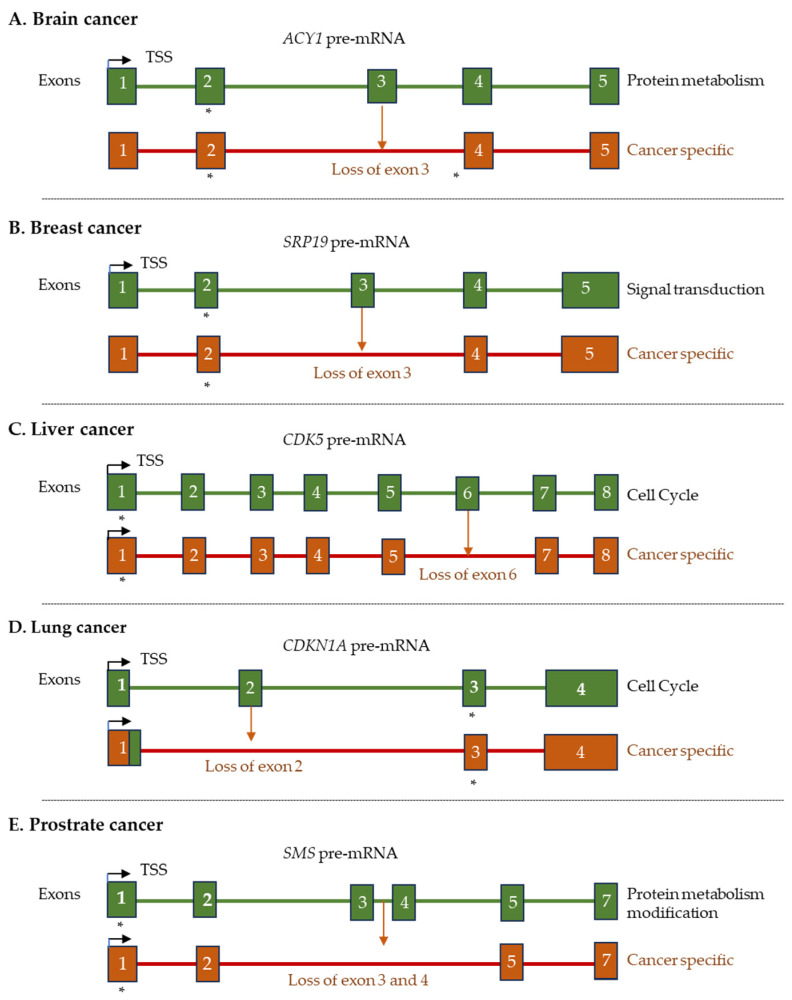
A schematic representation of cancer-specific alternative gene splicing. (**A**) Brain cancer (ACYl), (**B**) breast cancer (SRP19), (**C**) liver cancer (CDK5), (**D**) lung cancer (CDKN1A), and (**E**) prostate cancer (SMS). Cancer-specific transcript variants are shown on the bottom in each panel. The biological processes of these transcripts are indicated on the right. Deleted exons are shown with dark brown arrows, and start codons with * marks. The exons are numbered 1–8.

**Figure 5 ijms-26-09356-f005:**
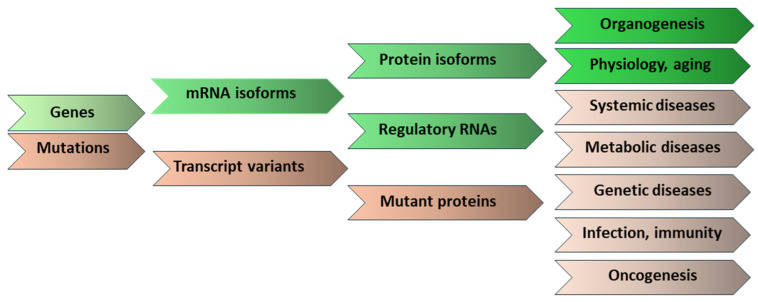
Gene expression, mRNA isoforms, and variants, proteins to complex phenotypes. A schematic presentation illustrating the variation at the gene, mRNA, and protein levels. The protein isoforms serve during organogenesis, normal functions, and aging, whereas the transcript variants may be responsible for disease pathogenesis or expressed in response to the disease state.

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
