# Peer review of "mRNA Isoforms and Variants in Health and Disease"

_ijms, 2025, doi:10.3390/ijms26199356_

Round 1

Reviewer 1 Report

Comments and Suggestions for Authors

This review summarizes the physiological and pathological roles of mRNA isoforms and transcript variants across development, organogenesis, aging, and a broad spectrum of diseases, including neurological, cardiovascular, respiratory, metabolic, genetic, autoimmune, infectious, and malignant disorders.

The manuscript is well structured, clearly written, and comprehensive in scope, making it potentially useful for readers in molecular biology and translational medicine. However, several major issues should be addressed before publication.

  1. In the section on physiological roles of mRNA isoforms (development, organogenesis, physiology, and aging), I recommend adding a summary table to improve clarity and readability. Such a table could list representative genes, key regulatory factors (e.g., transcription factors, splicing regulators, epigenetic modifiers), the specific isoform changes observed, and their associated functional outcomes. This would provide readers with a concise reference point, complementing the narrative text and highlighting the mechanistic links between isoform regulation and biological processes.
  2. In the current version of the review, the manuscript does not adequately address the cutting-edge technologies that have substantially driven isoform research in recent years—particularly single-cell transcriptomics and AI-integrated approaches. This omission weakens the review’s ability to present the field’s frontier. I strongly recommend adding a dedicated section that systematically introduces these latest methodological advances in isoform analysis—such as long-read sequencing in combination with single-cell RNA-seq and AI-powered splicing prediction tools—and supports the discussion with citations from high-impact publications (e.g., from Nature, Cell, Nature Communications, Cancer Discovery, 2022-2024). This addition would greatly reinforce the review’s novelty, comprehensiveness, and relevance to current research trends.
  3. In the disease-related sections, the manuscript currently cites classic examples such as FOS and RAF1, which are valuable. However, to enhance the review’s relevance and translational outlook, it should incorporate recent studies (within the last three years) demonstrating how isoforms originating from tumor immune evasion can serve as novel targets for therapies. Including such latest findings would substantially enrich the content and underscore the clinical impact of isoform research.

Moreover, in the Conclusion, it is essential to explicitly articulate how isoform research can contribute to disease diagnosis and prognosis prediction, and how it can inform precision therapy and drug development. Clearly stating these translational applications will significantly elevate the review’s practical and clinical relevance.

In the end this review provides a broad and clear overview of mRNA isoforms in physiology and disease, but it is largely descriptive and lacks critical depth. The manuscript underrepresents recent advances such as single-cell transcriptomics, and AI-based splicing prediction, and it lacks new findings on isoform variants in tumor immune evasion with therapeutic potential. And the conclusion should explicitly highlight how isoform research informs diagnosis, prognosis, and precision therapy. Overall, I recommend supplying the appropriately contents to improve novelty, critical analysis, and translational relevance.

Author Response

Response to Reviewer 1

Overall comments: This review summarizes the physiological and pathological roles of mRNA isoforms and transcript variants across development, organogenesis, aging, and a broad spectrum of diseases, including neurological, cardiovascular, respiratory, metabolic, genetic, autoimmune, infectious, and malignant disorders. The manuscript is well structured, clearly written, and comprehensive in scope, making it potentially useful for readers in molecular biology and translational medicine.

Responses to overall comments: We are thankful to the reviewer for her/ his comments. We have addressed the individual comments in the following section. The revised part of the manuscript is highlighted.

Major Comments

Query #1. Summary Table for Physiological Roles: To enhance clarity and readability in the section on physiological mRNA isoform roles (development, organogenesis, physiology, aging), include a summary table. The table should list: representative genes, Key regulatory factors (transcription factors, splicing regulators, epigenetic modifiers), specific isoform changes, and associated functional outcomes. Such a table provides a concise reference, complements narrative text, and clarifies mechanistic links.

Response to Query #1: We are thankful to the reviewer for the valuable comment. In our revised manuscript, we have included a summary table outlining the physiological roles, which will be included as a supplementary table (Table S2).

Query #2: Latest Methodological Advances: The manuscript does not sufficiently discuss cutting-edge technologies driving recent isoform research (2022–2024), especially single-cell transcriptomics, long-read sequencing at single-cell resolution, and AI-integrated tools for isoform/splicing analysis. Add a dedicated section systematically introducing these advances and cite high-impact recent publications (e.g., Nature, Cell, Nature Communications, Cancer Discovery) to reinforce novelty and comprehensiveness.

Response to Query #2: We agree with the reviewer’s suggestion. However, we would like to mention that we have recently published two manuscripts on the basic aspects of mRNA isoform research (PMID: 40149494 and 39940824). The article with PMID40149494 has described the advances in detection technologies (Vo et al, 2025. Detection of mRNA transcript variants, 2025, Genes, 16: 343). Therefore, we may not need to mention those techniques again in this manuscript.

Query #3: Tumor Immune Evasion and Therapeutic Targeting: Recent studies have illustrated the emerging role of transcript isoforms in tumor immune evasion and as therapeutic targets. Enrich the disease sections with findings from the last three years that demonstrate translational potential—highlighting the newest knowledge on isoform variants in tumor immune escape, which is of key clinical relevance.

Response to Query #3: We thank the reviewer for this suggestion. In our revised manuscript, we have included the recent studies that illustrate the emerging role of transcript isoforms in tumor immune evasion and their potential as therapeutic targets.

Minor Queries

Query #4: The manuscript’s conclusion does not sufficiently articulate the translational significance of isoform research. Please revise the concluding section to explicitly highlight how isoform analysis informs disease diagnosis, prognosis prediction, precision therapy, and drug development.

Response to Query #4: We agree with the reviewer’s insight and have made the necessary changes in our revised manuscript.

Query #5: Ensure that throughout the manuscript, appropriate recent examples and comparative references are supplied for all new methodological and biological advances discussed, to anchor the review in the current literature.

Response to Query #5: We acknowledge the reviewer's suggestion, and we have revised our manuscript to include the recent advances.

Query #6: Consider including summary tables or schematic illustrations where appropriate, to add clarity and aid reader comprehension, particularly when detailing gene examples, isoform functions, or technology overviews

Response to Query #6: We are thankful to the reviewer for the suggestion. According to the reviewer’s recommendation, we have incorporated a supplementary table (Table S1) in the revised manuscript.

Reviewer 2 Report

Comments and Suggestions for Authors

This is a well written review article about splicing isoforms for diseases.

It has very important and useful information for the readers about isoforms.

I have  few comments for this article.

1) It would be nicer if the authors could have a compilation table about splicing isoforms with the name of the genes and their functions.

2) It would be better to describe more detail about splicing mechanism with splicing factors to produce disease isoforms.

Author Response

Response to Reviewer 2

Overall comments

This is a well written review article about splicing isoforms for diseases. It has very important and useful information for the readers about isoforms.

Responses to overall comments: We are thankful to the reviewer for her/ his comments. We have addressed the individual comments in the following section. The revised part of the manuscript is highlighted.

Query #1: It would be nicer if the authors could have a compilation table about splicing isoforms with the names of the genes and their functions.

Response to Query #1: We thank the reviewer for the suggestion. We have included summary tables in the revised manuscript, indicating representative splicing isoforms, genes, and their functions (Supplementary Table S4).

Query #2: It would be better to describe more details about the splicing mechanism with splicing factors to produce disease isoforms.

Response to Query #2: We agree with the reviewer’s comment. We have prepared a summary table presenting representative splicing mechanisms, associated factors, and disease-specific variants (Supplementary Table S3).

Round 2

Reviewer 1 Report

Comments and Suggestions for Authors

Authors have addressed all my concerns.